# Preliminary Study of the Effect of Stereotactic Body Radiotherapy (SBRT) on the Immune System in Lung Cancer Patients Unfit for Surgery: Immunophenotyping Analysis

**DOI:** 10.3390/ijms19123963

**Published:** 2018-12-09

**Authors:** Arturo Navarro-Martín, Isabel Linares Galiana, Miguel A. Berenguer Frances, Jon Cacicedo, Rut Cañas Cortés, Silvia Comas Anton, Susana Padrones Sánchez, Santiago Bolívar Cuevas, Renate Parry, Ferran Guedea Edo

**Affiliations:** 1Radiation Oncology Department, Hospital Duran i Reynals, Institut Català d’Oncologia (ICO), Radiobiology and Cancer Group, ONCOBELL Program, Institut d’Investigació Biomèdica de Bellvitge (IDIBELL), Avinguda de la Gran Via de l’Hospitalet 199-203, L’Hospitalet de Llobregat, 08098 Barcelona, Spain; ilinaresgaliana@iconcologia.net (I.L.G.); maberenguer@iconcologia.net (M.A.B.F.); fguedea@iconcologia.net (F.G.E.); 2Radiation Oncology Department, Hospital Universitario de Cruces, Plaza de Cruces s/n, E-48903 Barakaldo, Bizkaia, Spain; jon.cacicedofernandezbobadilla@osakidetza.eus; 3Radiobiology and Cancer Group, ONCOBELL Program, Institut d’Investigació Biomèdica de Bellvitge (IDIBELL), Avinguda de la Gran Via de l’Hospitalet 199-203, L’Hospitalet de Llobregat, 08098 Barcelona, Spain; rcanas@idibell.cat; 4Radiation Oncology Department, Hospital Germans Trias i Pujol, 08098 Badalona, Spain; silvia.comas.anton@gmail.com; 5Respirology Department, Hospital Univesitari de Bellvitge, Hospitalet de Llobregat, Feixa Llarga s/n, E-08907 Barcelona, Spain; padrones@bellvitgehospital.cat; 6Radiology Department, Hospital Univesitari de Bellvitge, Hospitalet de Llobregat, Feixa Llarga s/n, E-08907 Barcelona, Spain; Sbolivar@bellvitgehospital.cat; 7Senior Director, Global Translational Science, Varian Medical Systems, 3120 Hansen Way, Palo Alto, CA 94304, USA; Renate.parry@varian.com

**Keywords:** stereotactic body radiotherapy (SBRT), immune system, immunomodulation, immunophenotyping, flow cytometry, non-small cell lung cancer, cytotoxic T lymphocytes, T helper lymphocytes, natural killer (NK) cells, regulatory T cells, myeloid-derived suppressor cells

## Abstract

An immunophenotyping analysis was performed in peripheral blood samples from seven patients with lung cancer unfit for surgery treated with stereotactic body radiotherapy (SBRT). The objective was to characterize the effect of SBRT on the host immune system. Four patients received 60 Gy (7.5 Gy × 8) and three 50 Gy (12.5 Gy × 4). Analyses were performed before SBRT, 72 h after SBRT, and at one, three, and six months after the end of SBRT. Of note, there was a specific increase of the immunoactive component of the immune system, with elevation of CD56^+high^CD16+ natural killer (NK) cells (0.95% at baseline to 1.38% at six months), and a decrease of the immunosuppressive component of the immune system, with decreases of CD4+CD25+Foxp3+CDA5RA− regulatory T cells (4.97% at baseline to 4.46% at six months), granulocytic myeloid-derived suppressor cells (G-MDSCs) (from 66.1% at baseline to 62.6% at six months) and monocytic (Mo-MDSCs) (8.2% at baseline to 6.2% at six months). These changes were already apparent at 72 h and persisted over six months. SBRT showed an effect on systemic immune cell populations, which is a relevant finding for supporting future combinations of SBRT with immunotherapy for treating lung cancer patients.

## 1. Introduction

Stereotactic body radiotherapy (SBRT) or stereotactic ablative body radiotherapy (SABR) has become the non-surgical treatment of choice for patients with non-small-cell lung cancer (NSCLC) who are deemed unfit for surgery [1]. In patients with operable stage I NSCLC, analysis of pooled data from phase 3 randomized trials comparing SBRT with surgery may tentatively support the notion that the two therapies are equally effective, without higher rates of regional metastases in patients treated with SBRT [2].

A rare clinical response to local radiotherapy is tumor regression outside the radiation field, commonly known as the abscopal effect. The term abscopal was coined by Mole [3] in 1953 from the latin “ab scopus”, i.e., away from the target. Experimental studies have suggested the key role of T lymphocytes as antitumor effectors in tumor response to radiation [4,5]. Furthermore, localized radiotherapy has been shown to induce abscopal effects in several types of cancer, including melanoma [6], metastatic NSCLC [7], lymphoma, and renal cell carcinoma [8,9]. Although the biologic characteristics underlying this effect are not well understood, it may be mediated by immunologic mechanisms [10]. The immunomodulatory effect of SBRT has been a matter of debate in different studies [11,12] in which a presumed synergistic effect of SBRT combined with immunotherapy has been hypothesized. The synergistic effect of both treatment modalities may improve clinical outcomes.

SBRT contributes to an antitumor immune response through multiple mechanisms, but a detailed understanding of the interactions of SBRT with the host immune system remains unclear. We assessed systemic immunity in primary and metastatic lung cancer patients unfit for surgery prior to and after SBRT to characterize the changes in immunophenotyping analysis. This information could be useful to optimize the combination and/or timing of future immunotherapeutic approaches with SBRT in cancer treatment.

## 2. Results

### 2.1. Clinical Characteristics of Patients

There were five men and two women, with a mean age of 73 years (range 65–80 years). Five patients were diagnosed with primary NSCLC, one patient with lung metastasis from colorectal cancer, and one patient with lung metastasis from breast cancer. Lung lesions were located in the right lower lobe in three patients, in the right upper lobe in two, and in the right middle lobe in two. All patients were deemed unfit for surgery after evaluation by the multidisciplinary tumor committee of our institution.

Four patients were treated with 7.5 Gy in 8 fractions (60 Gy) and three patients with 12.5 Gy in 4 fractions (50 Gy). After a median follow-up of 16 months (range 2–20 months), complete response of the target lesion was obtained in one patient, partial response in four, and stable disease in two. One of the patients with a stable target lesion developed systemic cancer progression.

### 2.2. Immunophenotyping Panel

Of the 35 samples obtained at different time intervals in the course of SBRT, 32 were analyzed. In one patient, a sample was lost at three months, and two patients did not complete the follow-up at six months.

In the gating strategy for immune cell types, lymphocytes were first screened on the basis of their side scattering and forward scattering characteristics, which were further examined for cell specific markers for identifying various immune cells. Flow cytometry diagrams for the analysis of all cell types are shown in Figure 1.

The prevalence of total lymphocytes in all seven patients before SBRT showed a median baseline value of 19.9%. The frequency of this immune cell type increased after the administration of radiotherapy, with a median of 29.1% at six months, and the highest values of 30.4% attained at three months of treatment.

Changes of total lymphocytes during the study period after the administration of SBRT showed a trend towards statistical significance using Friedman one-way ANOVA (*p* = 0.066). However, differences for the comparison of paired samples at different time points versus baseline, assessed with the Wilcoxon signed-rank test, were not statistically significant (*p* = 0.840) (Figure 2).

In relation to lymphocyte cell subpopulations, cytotoxic T cells (CD3+ and CD8+) showed an increase from baseline to 72 h of SBRT treatment, from 65% to 68%, although this increase was not statistically significant (*p* = 0.690), followed by a trend towards a decrease in this subpopulation up to 56.4% at 6 months. Changes of CD3+ and CD8+ cytotoxic T cells were statistically significant (Friedman’s test, *p* = 0.026), which indicates that this finding was relevant (Figure 3).

The subpopulation of T helper cells (CD3+CD4+) showed a progressive increase from 32.2% at baseline to 36.8% at six months of treatment, with a maximum peak of 43% at three months. These changes were not statistically significant neither using the Friedman’s test (*p* = 0.07) nor the Wilcoxon’s test for paired comparisons at different time points versus baseline (Figure 4).

The CD4+/CD8+ ratio showed a progressive increase from 0.50 at baseline to 0.65 at six months, with a maximum peak of 0.87 at three months.

The subpopulation of natural killer cells (NK) with activated phenotype, defined as CD56^+high^ CD16+ (hNK CD56+), showed an increase over the follow-up period from 0.95% at baseline to 1.38% at six months. These differences, however, were not statistically significant using the Friedman’s test (*p* = 0.180) nor the Wilcoxon’s test for paired comparisons at different time points versus baseline (Figure 5).

### 2.3. Regulatory T Cells (Treg cells)

The strategy used for the analysis of different phenotypic and functional Treg subsets was screening lymphocytes according to side scattering and forward scattering characteristics, which were additionally blocked for CD4+ T cells. Then, CD4+ T cells were examined for CD4+CD25+Foxp3+Helios cell populations. Tregs were also divided into functional subsets based on CD45RA and Foxp3 expression. The representative diagrams of flow cytometry for the analysis of all cell types are shown in Figure 6.

The time course of regulatory cells of activated phenotype (CD4+CD25+Foxp3+CD45RA−) and CD45RA− subset showed percentages of 4.97% and 4.46% at baseline and at six months, respectively (Figure 7).

### 2.4. Myeloid-Derived Supressor Cells (MDSCs)

Granulocytic MDSC (G-MDSC) were characterized as CD33+CD11b+CD14−, and monocytic MDSC (Mo-MDSC) were identified as CD33+CD11b+CD14+HLA-DR^−/low^. The representative diagrams of flow cytometry for the analysis of all cell types are shown in Figure 8.

Overall, there was a decrease of MDSCs after the use of SBRT. In the G-MDSC population, there was an increase at 72 h after SBRT administration, followed by a significant progressive decrease from 66.1% at baseline to 62.6% at six months (Friedman’s test, *p* = 0.01). In the Mo-MDSC population decreasing levels throughout the study period were observed, from 8.2% at baseline to 6.2% at six months, but differences were not statistically significant (Frieman’s test, *p* = 0.267). Both in the G-MDSC and Mo-MDSC populations, comparisons of median percentages at different time points versus baseline were not statistically significant (Figure 9).

## 3. Discussion

The effect of SBRT on the host immune system has received increasing attention in recent years, particularly exploring the ability of radiation to induce anti-tumor immune responses in certain specific settings, with anti-tumor T cells as key players in tumor control achieved by radiotherapy [4,13,14]. The present study provides novel evidence that SBRT has a systemic effect on the immune system, detectable in peripheral blood by an increase in activated NK lymphocytes (from 0.95% at baseline to 1.38% at six months) and a moderate decrease of Treg cells (from 4.97% at baseline to 4.46% at six months). Interestingly, these changes were already detected at 72 h after the administration of SBRT. In addition, the observation that MDSCs were those more extensively modified by radiotherapy over the study period, with a decrease in the immunosuppressive component, is a salient finding of the study.

A recent systematic review examined the correlation between radiation-associated lymphopenia and survival outcomes across various tumor types [15]. Although radiation therapy has an immunostimulatory effect via radiation-induced neoantigens and immune-activated danger signals [16], it also has immunosuppressive effects such as lymphopenia. This effect, however, depends on the timing of blood counts and time course of lymphopenia. On the other hand, studies included in the systematic review used normo-fractionated radiotherapy, and in one of the studies [17], the percentage of lung volume receiving low doses (5 Gy) had the highest association with lymphocyte nadir and worse outcomes. In our study, all patients received stereotactic radiotherapy with a very high dose gradient, so that the lung volume receiving low doses was negligible. Furthermore, the decrease of lymphocytes detected at 72 h after SBRT showed a subsequent recovery without significant differences at one, three, and six months as compared to baseline.

In relation to sized fractions of radiation dose, in a mouse melanoma model, Schaeu et al. [18] assessed how radiation dose and fraction size affected antitumor immunity. Fractionated treatment with medium-size radiation dose of 7.5 Gy/fraction was associated with the best tumor control and tumor immunity while maintaining regulatory T cells number. Three patients were treated with 7.5 Gy/fraction and 4 with 12.5 Gy/fraction, but the small study sample prevented to compare the response according to the size of dose per faction.

Increase of activated NK phenotype, defined by CD56^+high^CD16+, is related to high cytotoxic activity and low cytokine production [19], and its use in allogenic peripheral-blood stem cell transplantation has demonstrated that this T-cell subpopulation can induce tumor regression [20]. In our study, we found an increase from 0.95% at baseline to 1.38% at six months. This effect may explain, in part, the favorable results of SBRT consistently reported in the literature [21]. Radiation produces upregulation of cell surface proteins as tumor-associated antigens or major histocompatibility (MHC) molecules enhance the activity of activated lymphocytes [22]. This phenomenon increases cross presentation and cell trafficking in peripheral blood [13]. Chen et al. [23] described a series of stepwise events as the cancer-immunity cycle, which characterizes the anticancer immune response leading to the effective killing of cancer cells. After SBRT, it may be hypothesized that we have increased the priming and activation of effector T cell responses, which in turn could be a reason for the increase of activated NK lymphocytes seen at follow-up. 

Suppressor phenotype of regulatory T cells (CD4+CD25+Foxp3+CD45RA−) was determined according to previous studies [24]. The frequency of circulation of Treg subpopulations with immunosuppressive activity has been associated with poor patient survival in lung cancer among other solid tumors [25]. In our study, regulatory T cells of the activated phenotype showed a systemic decrease from baseline values to values at six months (4.97% vs. 4.46%), which may represent a therapeutic advantage for these type of tumors treated with SBRT.

Cancer-induced MDSCs play an important role in tumor immune evasion mechanisms [26]. Recently, an interesting study by Jayaraman et al. [27] showed that MDSC induced in the presence of tumor growth factor (TGF)-β1 cytokine, particularly in combination with radiotherapy, acquired a novel phenotype characterized by loss of T cell suppression ability and acquisition of enhanced antigen-presenting and tumor killing functions. These findings support the potential benefit of leveraging the plasticity of MDSCs for therapeutic benefit. Mo-MDSC inhibit T cell responses through nitric oxide (NO)-related pathways resulting in production of NO and the upregulation of PD-L1 expression. In reference to G-MDSC (CD33−CD11b+CD14−), studies of small-cell lung cancer cells [28] have related irradiation with secretion of granulocyte-macrophage colony-stimulating factor (GM-CSF) by fibroblasts and tumor cells accelerating tumor invasion but not growth. Results of the present study showed a decrease of both G-MDSC and Mo-MDSC during the study period providing evidence of a reduction of the immunosuppressive component of MDSCs elicited by SBRT.

Even though some studies have found that radiation induced stimulation of cytokine granulocyte-macrophage colony-stimulating factor [29,30], all these studies have been tested with large fields as are used in conventional radiation therapy. However, SBTR is a technique which allows the delivery of high doses in small fields, so it is unclear whether radiation delivered by SBRT can produce an increase of granulocyte levels.

### Limitations of the Study

A limitation of the study is its preliminary exploratory nature and the fact that the effect of SBRT on the host immune system was evaluated in a reduced number of seven lung cancer patients. Therefore, the present findings should be cautiously interpreted. It should be noted that this was a translational substudy of patients included in a prospective phase 2 clinical trial to assess the safety of SBRT in selected patients with stage I NSCLC and metastatic lung cancer; therefore, the patients were treated with all the guarantees of a clinical trial in terms of homogeneity, follow-up, and safety. 

## 4. Materials and Methods

### 4.1. Design and Patients

Seven patients with primary NSCLC or metastatic lung cancer who met the criteria for our ongoing phase 2 clinical trial (registered at https://clinicaltrials.gov/ ClinicalTrials.gov identifier NCT01823003) and who consented were enrolled in this translational substudy. Briefly, the phase 2 trial was a prospective, interventional, open-label, non-randomized, and single-center study, which was designed to evaluate the safety and feasibility of SBRT in selected patients with NSCLC or metastatic lung cancer. The objective of the present substudy was to assess the systemic response of the immune system elicited by SBRT.

The multidisciplinary thoracic tumor board of our hospital approved the treatment and established the criteria of medical inoperability of the patients. The study protocol was approved by the Institutional Review Board (code sbrt_lung_fff2012; code AC026/12, approval in February 2018). Written informed consent was obtained from all participants.

### 4.2. Eligibility

Inclusion criteria were histologically-confirmed primary lung cancer or lung metastasis originating from another primary tumor, tumor diameter <5 cm, medically inoperable patients or medically operable patients who refused surgery, life expectancy > 12 months, age > 18 years, Barthel score > 40, Karnofsky performance status (KPS) > 70, and ability to understand and willingness to sign a written informed consent. Criteria for inoperability included overall assessment by the institutional thoracic tumor committee and reduced pulmonary function based on one major or two minor criteria (major criteria: forced expiratory volume in one second (FEV_1)_ < 50% predicted or < 1 L and diffusing capacity for carbon monoxide [DLCO] < 50%; minor criteria: age > 75 years, FVE_1_ 51–60% predicted or 1–1.2 L, DLCO 51–60%, pulmonary hypertension, left ventricular ejection fraction ≤ 40%, resting or exercise arterial pO_2_ < 55 mm Hg, and pCO_2_ > 45 mm). All patients underwent a positron-emission tomography (PET)-CT scan within two months before radiotherapy. 

### 4.3. SBRT Treatment

Details of the SBRT technique have been previously described [29]. All patients received SBRT according to localization and size of the pulmonary lesions following the study protocol of the phase 2 clinical trial and fulfilling all the inclusion criteria. Briefly, SBRT was delivered using volumetric modulated arcs with photon beam energy of 6–10 MV. The number of arcs and their ballistic was left undefined in order to allow for optimization of dose distribution. Bolus was not allowed. The protocols of radiation were as follows: (a) 34 Gy in a single fraction (distance to chest wall > 1 cm, tumor size < 2 cm, and distance to the main bronchus > 2 cm; (b) 54 Gy (18 Gy in 3 fractions) (distance to chest wall > 1 cm, tumor size between 2 and 5 cm, and distance to the main bronchus > 2 cm); (c) 60 Gy (12 Gy in 5 fractions) (distance to chest wall < 1 cm, tumor size < 5 cm, and distance to main bronchus > 2 cm); and (d) 60 Gy (7.5 Gy in 8 fractions (tumor size < 5 cm and distance to the main bronchus < 2 cm). 

Response of the primary lesion was evaluated using the Response Evaluation Criteria for Solid Tumors (RECIST) [31] at 1, 3, 6, and 12 months after SBRT.

### 4.4. Blood Samples

Peripheral blood samples were obtained in heparinized tubes before SBRT, 72 h after SBRT, and at 1, 3, and 6 months after the end of treatment.

Fresh blood was used and samples were processed in less than 24 h after extraction, so that the cell viability procedure could be omitted.

Peripheral blood mononuclear cells (PBMCs) were isolated from a heparinized venous blood sample by density gradient centrifugation. The blood was diluted 1:1 with saline before being layered onto Ficoll^®^ Plaque Plus (GE Healthcare Bio-Sciences, Pittsburgh, PA, USA). After centrifugation, PBMCs were collected from the plasma-Ficoll interphase and used for flow cytometry assays.

### 4.5. Flow Cytometry

Premixed DuraClone IM Antibody Panels^®^ were used for flow cytometry analyses. The panels of interest were the following:-Lymphocyte Phenotyping DuraClone^TM^, Beckman Coulter Life Sciences (Indianapolis, IN, USA): CD16 antibody (Ab), CD56 Ab, CD19 Ab, CD14 Ab, CD4 Ab, CD8 Ab, CD3 Ab, and CD45 Ab.-Regulatory T Cells DuraClone^TM^, Beckman Coulter Life Sciences: CD45RA Ab, CD25 Ab, CD39 Ab, CD4 Ab, intracellular Foxp3 Ab, CD3 Ab, and CD45 Ab.-Myeloid-derived Suppressor Cells (MDSC) DuraClone^TM^, Beckman Coulter Life Sciences: CD45, HLA-DR, CD14, CD33, and CD11b.

Cell surface and intracellular staining were performed following the manufacturer’s protocols. A minimum number of 100,000 events for lymphocyte phenotyping and MDSC analyses and 40,000 events for regulatory T cells analysis was established [32]. Cell phenotypes were evaluated using the FACS Navio system (Beckman Coulter Life Sciences). Data were analyzed with the FlowJo LLC software (Tree Star Inc., Ashland, OR, USA).

### 4.6. Statistical Analysis

Categorical data are expressed as frequencies and percentages, and continuous data as median and range. The Friedman one-way ANOVA was used to compare changes of cell populations at different time periods, and the Wilcoxon signed-rank test to assess differences of cell populations at different time periods as compared with baseline. Statistical significance was set at *p* < 0.05. Data were analyzed using the Statistical Package for the Social Sciences (IBM Corp., SPSS for Windows version 20.0, Armonk, NY, USA).

## 5. Conclusions

In this preliminary study of patients who receive SBRT for lung tumors, some changes of circulating blood immune cell populations were observed. These changes failed to reach statistical significance, but included an increase of the immunoactive component of the immune system with elevation of CD56+highCD16+ NK-cells, and a decrease of immunosuppressive component with decreases of CD4+CD25+Foxp3+CD45RA− regulatory T cells and G-MDSC and Mo-MDSC. Notably, these immune responses were already apparent as early as 72 h after SBRT administration and persisted over six months after treatment. It is important to note that the non-significant results are impacted by the small number of samples. Further studies with number of samples and participants based on power calculations are needed to examine the effect of SBRT on the host immune system. Knowledge of the immune responses elicited by SBRT may help to define the potential role of the combination of SBRT with immunotherapy, such as adoptive immunotherapy and/or checkpoint inhibition as future therapeutic options in lung cancer patients.

## Figures and Tables

**Figure 1 ijms-19-03963-f001:**
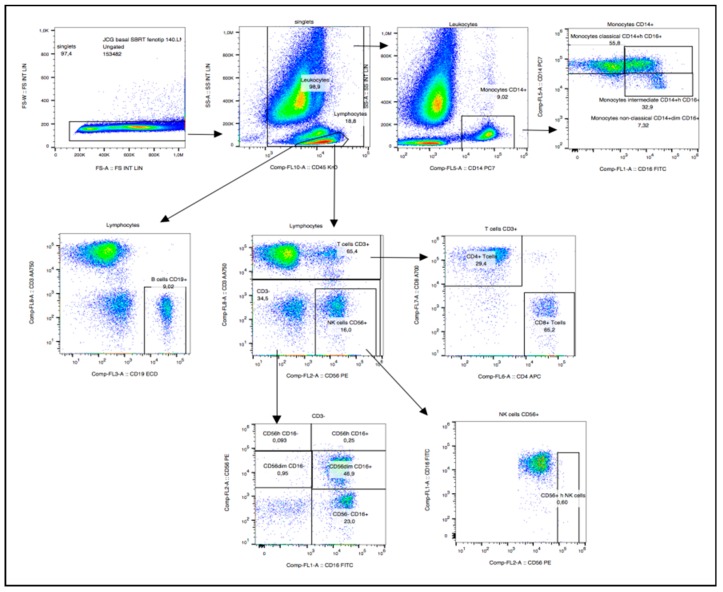
Immunophenotyping panel.

**Figure 2 ijms-19-03963-f002:**
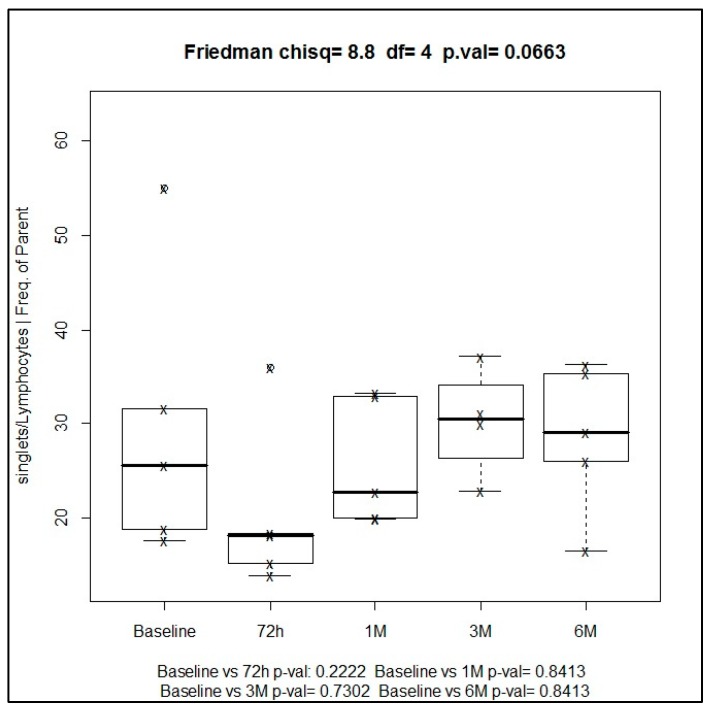
Median values of total lymphocytes during the study at different time points.

**Figure 3 ijms-19-03963-f003:**
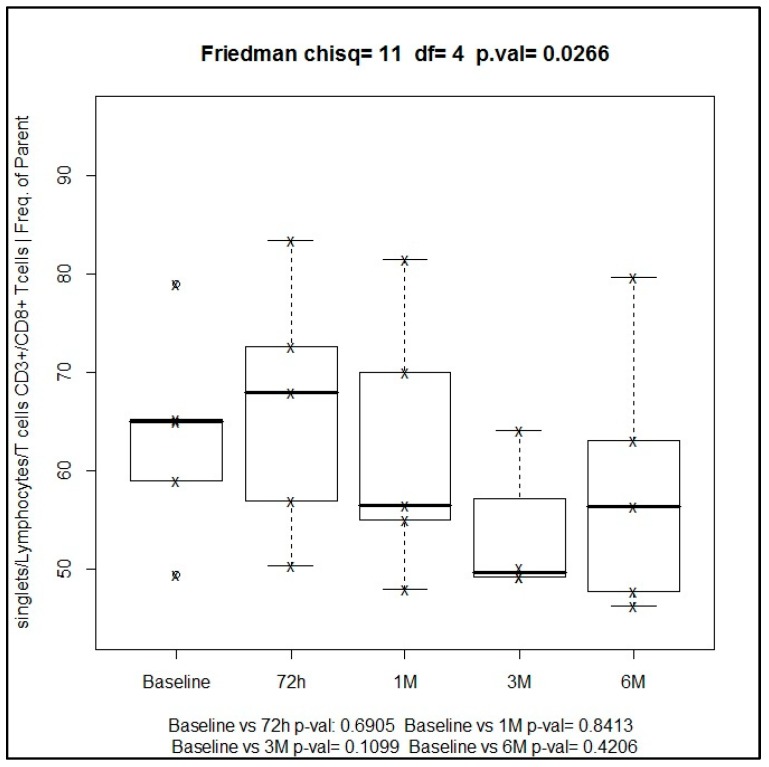
Median values of cytotoxic T cells (CD3+ and CD8+) during the study at different time points.

**Figure 4 ijms-19-03963-f004:**
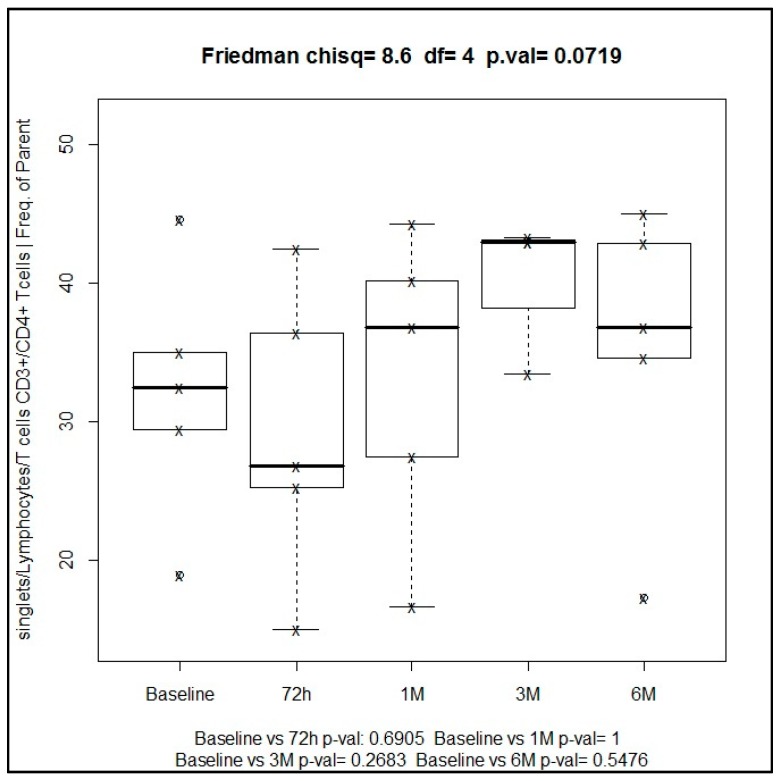
Median of percentage values of T helper cells (CD3+CD4+) at different time points.

**Figure 5 ijms-19-03963-f005:**
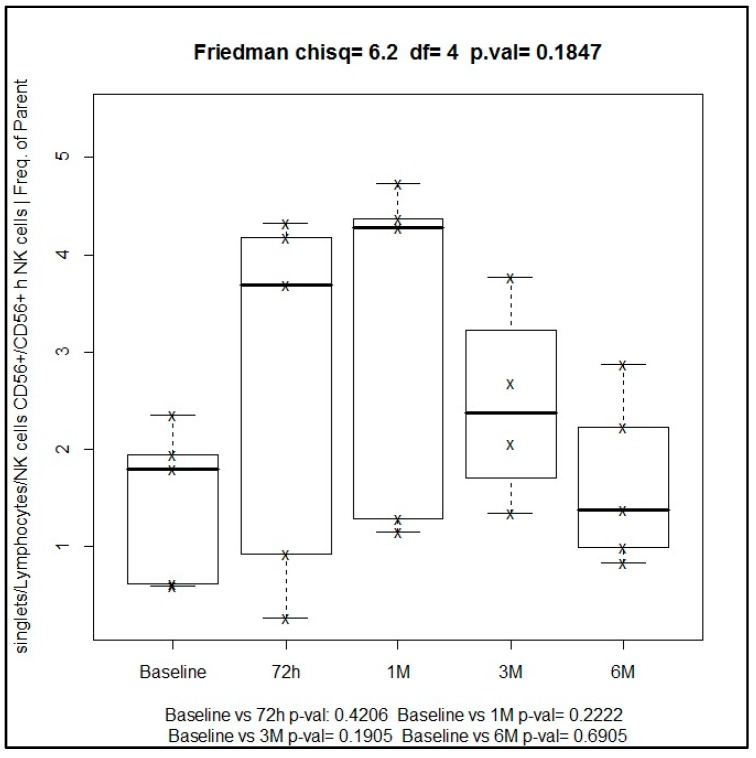
Median percentage of activated NK cells (hNK CD56+) during the study period.

**Figure 6 ijms-19-03963-f006:**
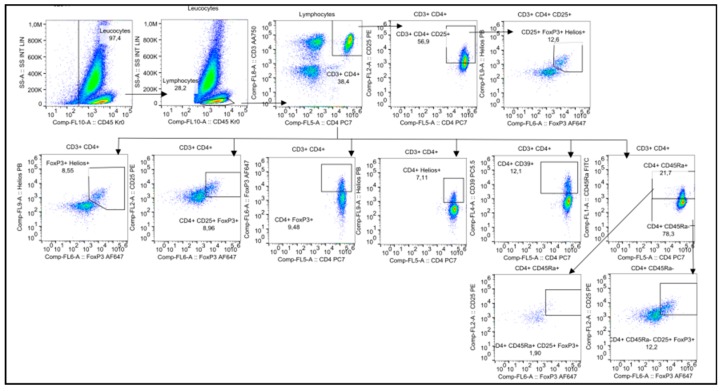
Immunotyping panel for regulatory T cells.

**Figure 7 ijms-19-03963-f007:**
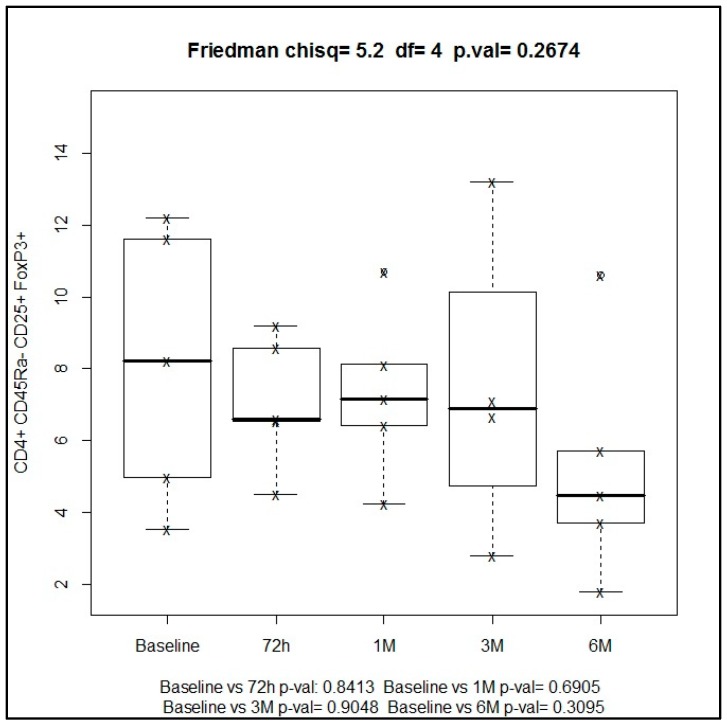
Median percentage of regulatory T cells during the study period at different time points.

**Figure 8 ijms-19-03963-f008:**
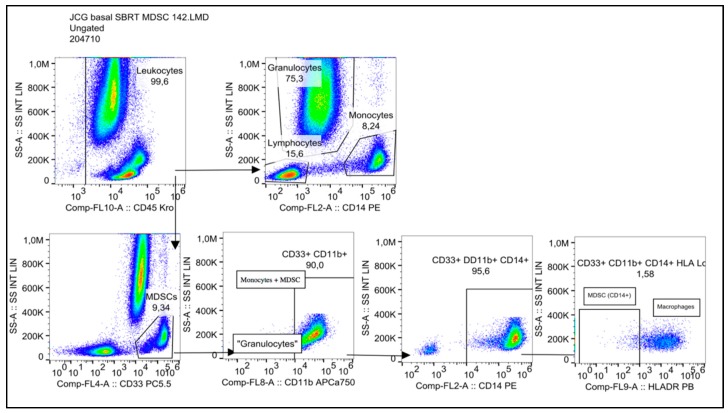
Immunotyping panel for myeloid-derived suppressor cells (MDSCs).

**Figure 9 ijms-19-03963-f009:**
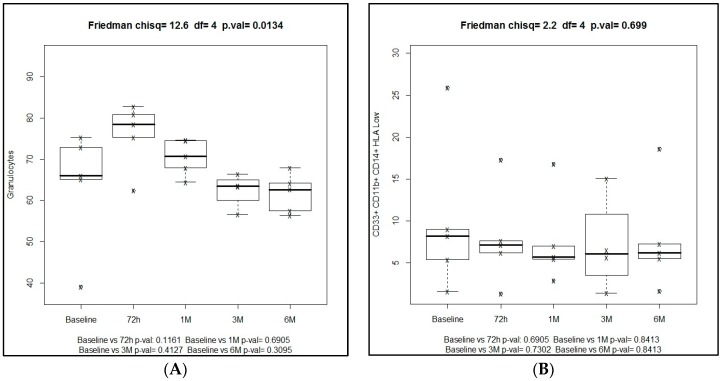
(**A**) Box plots of granulocytic MDSC (G-MDSC) CD33+CD11b+CD14− and (**B**) Monocytic MDSC (Mo-MDSC) CD33+CD11b+CD14+HLA-DR^−/low^ values at different time points of the study.

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
