# Peer review of "Preliminary Study of the Effect of Stereotactic Body Radiotherapy (SBRT) on the Immune System in Lung Cancer Patients Unfit for Surgery: Immunophenotyping Analysis"

_ijms, 2018, doi:10.3390/ijms19123963_

Round 1

Reviewer 1 Report

The conclusion doesn’t correlate with the findings. The increase in number of cytotoxic T cells was not statistically significant. Most of the other immune cell counts failed to show significant increase. The conclusion needs to reflect the negative results of the study.

Even though this study looks into an interesting question for the abscopal effects of SABR treatment, the number of samples and participants wasn’t based on a power calculation. The negative results are impacted by the small number of samples.

Author Response

The authors would like to thank the Reviewer for his comments, which have contributed to clarify and improve our reporting, in particular to tone down the conclusions.

The conclusion doesn’t correlate with the findings. The increase in number of cytotoxic T cells was not statistically significant. Most of the other immune cell counts failed to show significant increase. The conclusion needs to reflect the negative results of the study.

Given the characteristics of the study and the small sample size, we have included “Preliminary study ….” in the title.

We have written a more cautious conclusion based on the lack of statistical significance in the increase of cytotoxic T cells and of other immune cell counts. The new conclusion reads as follows: “In this preliminary study of patients who receive SBRT for lung tumors, some changes of circulating blood immune cells populations were observed. These changes failed to reach statistical significance, but included an increase of the immunoactive component of the immune system with elevation of CD56+highCD16+ NK-cells, and a decrease of immunosuppressive component with decreases of CD4+CD25+Foxp3+CD45RA- regulatory T cells and G-MDSC and Mo-MDSC.”

Even though this study looks into an interesting question for the abscopal effects of SABR treatment, the number of samples and participants wasn’t based on a power calculation. The negative results are impacted by the small number of samples

This point is now also reflected in the conclusion with these new sentences: “It is important to note that the non-significant results are impacted by the small number of samples. Further studies with number of samples and participants based on power calculations are needed to examine the effect of SBRT on the host immune system.”

Reviewer 2 Report

The authors present a clinical report of the effect of SBRT on immune system. The majority of the report is based on immnunophenotyping panel using flow cytometry after SBRT treatment at indicated time point.. The data set is fairly small due to the fact that only 7 patients are invovled in this study. How SBRT affects the immune system is a significant topic of unknown mechanism. However, this manuscript does not answer this important question although the authors pointed out that it is related to this topic. Thus overall this work lacks of mechanistic insight into the topic they claimed to study. I wondering whether the author can clarify why they observe such changes in immunophenotyping analysis after SBRT treatment. The paper may be acceptable if more details on the molecular mechanisms are present. 

Author Response

We appreciate your suggestions regarding a more extensive comment of the molecular mechanisms underlying the effect of SBRT on the immune cells populations. This new information gives more scientific consistency to the discussion.

The authors present a clinical report of the effect of SBRT on immune system. The majority of the report is based on immunophenotyping panel using flow cytometry after SBRT treatment at indicated time point. The data set is fairly small due to the fact that only 7 patients are involved in this study.

The implications of the small sample size in the study findings were also noted by the other Reviewer. Besides a specific mention regarding this point in the Limitations of the Study section, we have added this sentence in the Conclusions: “It is important to note that the non-significant results are impacted by the small number of samples. Further studies with number of samples and participants based on power calculations are needed to examine the effect of SBRT on the host immune system.”

How SBRT affects the immune system is a significant topic of unknown mechanism.

We agree, and this was already stated in the last paragraph of the introduction: “SBRT contributes to an antitumor immune response through multiple mechanisms, but a detailed understanding of the interactions of SBRT with the host immune system remains unclear.”

However, this manuscript does not answer this important question although the authors pointed out that it is related to this topic. Thus overall this work lacks of mechanistic insight into the topic they claimed to study. I wondering whether the author can clarify why they observe such changes in immunophenotyping analysis after SBRT treatment. The paper may be acceptable if more details on the molecular mechanisms are present.

We have added further data in the discussion and new supporting references. The two new paragraphs are the following:

“Radiation produces upregulation of cell surface proteins as tumor-associated antigens or major histocompatibility (MHC) molecules enhancing activity of activated lymphocytes [22]. This phenomenon increases cross presentation and cell trafficking in peripheral blood [23]. Chen et al. [24] described a series of stepwise events as the cancer-immunity cycle, which characterizes the anticancer immune response leading to effective killing of cancer cells. After SBRT, it may be hypothesized that we have increased the priming and activation of effector T cell responses, which in turn could be a reason for the increase of activated NK lymphocytes seen at follow-up.”

“Even though some studies have found that radiation induced stimulation of cytokine granulocyte-macrophage colony-stimulating factor [30,31], all these studies have been tested with large fields as we use in conventional radiation therapy. However, SBTR is a technique which allows us to deliver high doses in small fields, so it is unclear whether radiation delivered by SBRT can produce an increase of granulocyte levels.”

New references include:

#22 Garnett CT, et al. Sublethal irradiation of human tumor cells modulates phenotype resulting in enhanced killing by cytotoxic T lymphocytes. Cancer Res 2014;64:7985-94.

#23 Park B, et al. The effect of radiation on the immune response to cancers. Int J Mol Sci 2014;15;927-43.

#24 Chen DS et al. Oncology meets immunology: the cancer-immunity cycle. Immunity 2013;39:1-10.

#30 Vilalta M, et al. Recruitment of circulating breast cancer cells is stimulated by radiotherapy. Cell Rep 2014;8:402-9.

#31 Vilalta M, et al. The role of granulocyte macrophage stimulating factor (GM-CSF) in radiation-induced tumor cell migration. Clin Exp Metastasis 2018;35:247-54.

Round 2

Reviewer 2 Report

The authors addressed the concerns I raised during the last round of review. I support its publication in the journal due to its improved quality.